# Formation of Transient Protein Aggregate-like Centers Is a General Strategy Postponing Degradation of Misfolded Intermediates

**DOI:** 10.3390/ijms241311202

**Published:** 2023-07-07

**Authors:** Susanna Boronat, Margarita Cabrera, Montserrat Vega, Jorge Alcalá, Silvia Salas-Pino, Rafael R. Daga, José Ayté, Elena Hidalgo

**Affiliations:** 1Oxidative Stress and Cell Cycle Group, Universitat Pompeu Fabra, C/Doctor Aiguader 88, 08003 Barcelona, Spain; susanna.boronat@upf.edu (S.B.); mcabrera@cbm.csic.es (M.C.); montserrat.vega@upf.edu (M.V.); alcalarumbe@gmail.com (J.A.); jose.ayte@upf.edu (J.A.); 2Centro de Biología Molecular Severo Ochoa and Departamento de Biología Molecular, Universidad Autónoma de Madrid (UAM), 28049 Madrid, Spain; 3Centro Andaluz de Biología del Desarrollo, Universidad Pablo de Olavide-Consejo Superior de Investigaciones Científicas-Junta de Andalucía, Carretera de Utrera, km1, 41013 Seville, Spain; ssalpin@upo.es (S.S.-P.); rroddag@upo.es (R.R.D.)

**Keywords:** heat shock response, protein aggregation-like centers, budding yeast, W303 vs. BY4741, fission yeast, Mas5, Ydj1, Btn2, Hsp42, Hsp104

## Abstract

When misfolded intermediates accumulate during heat shock, the protein quality control system promotes cellular adaptation strategies. In *Schizosaccharomyces pombe*, thermo-sensitive proteins assemble upon stress into protein aggregate-like centers, PACs, to escape from degradation. The role of this protein deposition strategy has been elusive due to the use of different model systems and reporters, and to the addition of artificial inhibitors, which made interpretation of the results difficult. Here, we compare fission and budding yeast model systems, expressing the same misfolding reporters in experiments lacking proteasome or translation inhibitors. We demonstrate that mild heat shock triggers reversible PAC formation, with the collapse of both reporters and chaperones in a process largely mediated by chaperones. This assembly postpones proteasomal degradation of the misfolding reporters, and their Hsp104-dependent disassembly occurs during stress recovery. Severe heat shock induces formation of cytosolic PACs, but also of nuclear structures resembling nucleolar rings, NuRs, presumably to halt nuclear functions. Our study demonstrates that these distantly related yeasts use very similar strategies to adapt and survive to mild and severe heat shock and that aggregate-like formation is a general cellular scheme to postpone protein degradation and facilitate exit from stress.

## 1. Introduction

The maintenance of a correctly folded proteome, or proteostasis, relies on the protein quality control (PQC) system. It includes the regulation of protein translation, chaperone-assisted protein folding and protein degradation functions (for reviews, see [1,2,3]). Proteostasis can be challenged by enhanced protein synthesis, by heat shock or other stresses or by age- or disease-related inactivation of PQC components. Then, the adjustment of the PQC strategies to the new demands will be essential for adaptation and survival (for reviews, see [4,5]). These new strategies often invoke the activation of signaling cascades and downstream transcriptional triggering of anti-stress responses. Thus, activation of the heat shock factor Hsf1 or of general stress-response pathways contribute to cell survival upon heat shock imposition in most cell types (for reviews, see [6,7,8]). Some of the genes up-regulated by these activated cascades code for chaperones, proteasome components or autophagy-related proteins. The upgraded folding and degradation machineries would then counterbalance the accumulated misfolded polypeptides (for reviews, see [8,9,10,11,12,13]).

Another important strategy to prevent and combat the accumulation of misfolded intermediates is the spatial sequestration of non-native intermediates into discrete foci (for reviews, see [14,15,16,17]). This sequestration is a protective strategy against heat and other stresses shared by bacteria and eukaryotes, but the localization differs between cell types. The use of temperature-sensitive alleles fused to fluorescent tags has allowed the visualization of protein aggregate-like foci in different model systems, many of which are formed when the degradative cellular functions, mainly components of the ubiquitin–proteasome system (UPS), are impaired. The formation and role of these foci are a matter of debate, as well as the terminology used to describe them. An extensively used organism to study PQC and protein aggregation is *Saccharomyces cerevisiae.* In this model system, the discovery of some aggregation-like centers, termed juxtanuclear quality control compartment (JUNQ) and insoluble protein deposit (IPOD), when the cell was subjected to heat shock and proteasome inhibition was pioneer in the field of PQC [14,18,19]. They were originally proposed to be transient storage centers for misfolded proteins which had to be degraded, with the idea that the spatial sequestration of the aggregating protein intermediates into these deposits would separate them from the healthy native proteome and would limit their spreading. An updated review of the literature reporting heat shock-dependent foci formation in *S. cerevisiae* indicates that there are three major sequestration sites for misfolded proteins: IPOD, INQ (intranuclear quality control compartment [19,20], probably former JUNQ but described as displaying nuclear localization) and CytoQ (quality control bodies [19]; probably former Q-bodies [18,21]). While IPOD mainly contain amyloidogenic proteins and terminally unfolded polypeptides, the nuclear INQ and CytoQ compartments include misfolded intermediates and require aggregases such as Btn2 and Hsp42 for deposition [19]. The proposed functions of these foci are multiple: regulating protein function [22], shielding other aggregation-prone proteins from misfolding, avoiding the collapse of PQC components during acute stress, or preparing polypeptides for either degradation or refolding.

In fission yeast, we have recently proposed that an important function for the spatial sequestration of misfolded proteins during heat stress is to avoid their degradation, so that the transiently misfolded intermediates can be refolded and reactivated during stress recovery [23]. Thus, during both mild and severe heat shock, the Hsp40 Mas5 drives the sequestration of misfolded intermediates into discrete foci, which we term protein aggregate centers (PACs), and which most likely correspond to budding yeast’s INQ plus CytoQ. PACs help misfolded polypeptides escape from degradation, and allow for their future refolding once physiological temperatures return; PAC disassembly depends on the disaggregase Hsp104 [23]. The relevance of this PAC-mediated strategy, escaping from degradation, is particularly highlighted during recovery experiments, that is, concluding heat shock by decreasing the temperature to physiological levels. The studies with budding yeast are very limited in this regard. Thus, protein refolding has been monitored in budding yeast after mild heat shock but using genetic or chemical proteasome inhibition conditions to enhance foci accumulation prior to recovery [21,24,25]; these artificial interventions hinder the interpretation of results (see next paragraph). A report clearly stating a role for aggregation-like centers in protecting proteins from degradation to promote cellular recovery is the work by Wallace, Drummond and colleagues [26], which demonstrated that some heat shock-dependent foci could be dissolved and proteins could be refolded during growth resumption; importantly, the heat shock conditions were highly toxic, and mild stress was not tested. Similarly, severe stress was used to study nuclear protein clearance after foci formation [27,28]. In the same line, it has also been proposed in *S. cerevisiae* that several metabolic and regulatory proteins required for cell cycle progression contain prone-to-aggregate domains, and reversibly coalescence into foci during stress to promote and regulate cell growth and survival [29].

In budding yeast, the localization, timing, shape and requirements of aggregation-like centers are very different depending on the study, and consequently, different names and functions have been assigned to the collapsed foci. Some of these differences may be specific to the stress type or to the severity of the heat shock, may depend on the fluorescent misfolding reporter used, or may even vary depending on the cell type. Additionally, researchers often use inhibitors of translation such as cycloheximide (CHX) or inhibitors of the proteasome such as MG132 or bortezomib during the course of their experiments, and these interventions alter the shape and dynamics of the deposition sites. In the case of the translation inhibitor CHX, its addition fully blocks PAC formation [23,30,31]. Regarding UPS inhibitors, even though they do not seem to affect the fundamental rules governing the formation and sorting of foci [25], the addition of MG132 or bortezomib was originally used to describe the formation of JUNQ and IPOD, which were augmented by the inhibitors [18,21]. Finally, even within the same model organisms, the use of different genetic backgrounds can alter the dynamics of aggregation-like foci. Thus, while *S. cerevisiae* has been extensively used to study heat shock responses at the genetic and cell biology levels, the protein aggregate deposits and the sorting factors involved in their formation may vary between the main two backgrounds used by the scientific community, namely W303 and BY4741 derivatives.

In view of all of the above and to compare whether both model systems, fission and budding yeasts, share PAC formation as a beneficial strategy during heat shock to transiently escape from degradation, we expressed the same misfolding reporters in both yeasts. We studied the formation and function of these aggregation-like foci during mild and severe heat shock in the absence of added inhibitors. We used both *S. cerevisiae* genetic backgrounds and two misfolded reporters (Rho1.C17R and Guk1-9), each deriving from endogenous fission yeast or budding yeast proteins, to rule out protein-specific fates. We demonstrate that PACs are formed in both W303 and BY4741 derivatives at mild and severe heat shock without the addition of proteasome inhibitors, PAC formation protecting the misfolded proteins from degradation; PAC kinetics and genetic requirements are slightly different in these *S. cerevisiae* backgrounds. Our biochemical analysis shows that the steady-state levels of misfolded substrates are well preserved 1–2 h after mild heat shock, allowing their refolding if permissive temperatures are recovered. We conclude that PAC formation is a general strategy used by these distant yeasts, and probably by many eukaryotes, to postpone protein degradation and promote cell survival in response to heat stress.

## 2. Results

### 2.1. Rho1.C17R-GFP Collapses into Protein Aggregate-like Centers (PACs) upon Mild Heat Stress in Budding Yeast

We expressed in budding yeast Rho1.C17R-GFP a misfolding reporter previously used in *S. pombe* [23]. Growth temperatures (25 °C or 30 °C) were adjusted differently to obtain comparable reporter levels in both yeast models. As shown in Figure 1a, Rho1.C17R-GFP displayed dual cytosolic and nuclear localization at the permissive temperatures in both yeasts. Similar to fission yeast, while growth rates of *S. cerevisiae* cultures during the logarithmic phase did not differ during growth at mild heat shock (37 °C) compared to the permissive temperature, growth of budding yeast was permanently halted after a shift to 42 °C (Figure 1b).

We analyzed the kinetics of foci formation upon mild heat shock in fission yeast and in two budding yeast backgrounds, namely BY4741 (directly arising from the original S288C) and W303 (also derived from S288C but including crosses to several other strains; [32]). Several reports have demonstrated significant differences between these two backgrounds regarding mitochondrial genome instability [33], oxidative stress signaling [34], salt tolerance [35] or replicative life span [36], among others. As described before in fission yeast, Rho1.C17R-GFP expressed in both *S. cerevisiae* backgrounds displayed dual cytoplasmic and nuclear localization at permissive temperature (Figure 1c; 25 °C/30 °C). Using the co-staining of Rho1.C17R-mCherry with nuclear markers (Appendix A), we confirmed that the slightly stronger staining observed in both budding yeast backgrounds coincided with the nucleus; as will be seen later, this facilitated the identification of nuclear localization events. The localization of Rho1.C17R-GFP after a shift to 37 °C changed from its homogeneous cytosolic/nuclear distribution to the formation of foci with different dynamics, being faster and maximal in both strains of *S. cerevisiae* than in *S. pombe*. The diffuse localization of wild-type Rho1 was unaffected during mild heat stress, demonstrating that foci formation is a consequence of protein misfolding (Appendix A). In the W303 background, PACs only lasted 30 min (Figure 1c, Rho1.C17R-mCherry in Appendix A), while they were more stable in BY4741 and even more in *S. pombe* (Figure 1c). The same was observed using another misfolding reporter, Guk1-9-GFP (Appendix A) [37]. When the stability of the reporter was analyzed by Western blot in the absence of inhibitors, the protein levels remained intact two hours after heat shock imposition in extracts from fission yeast and BY4741, but the concentration was clearly declined in extracts from the W303 background (Figure 1d,e). The proportionality between the duration of the foci and the protein stability in the three cell types suggests that PACs constitute a barrier for the degradation of misfolded polypeptides.

To confirm that the composition of these PACs in *S. cerevisiae* included chaperones, we monitored the localization of some classical ‘suspects’ using GFP-tagged versions of proteins reported to be present at IPOD, JUNQ/INQ or CytoQ [18,19,25]. In this case, the proteins were analyzed in the BY4741 background, where the collection of GFP derivatives had been constructed [38]. It is worth mentioning that the strain carrying the *hsp104-GFP* allele, frequently used in the field, displayed some growth defects upon heat shock (Appendix A). As shown in Figure 2 and Appendix A, Btn2, Hsp42, Hsp104, Hsp26 and Sse1 (38) collapsed into foci upon mild heat shock, with similar kinetics to the misfolding reporters, confirming that PAC formation may be a cellular strategy to circumvent stress. In agreement with earlier studies [19,39], the signal of the chaperone Btn2 decreased faster than Hsp42-GFP and Hsp104-GFP during mild heat stress, probably due to its lower expression (Hsf1-dependent) and rapid degradation.

### 2.2. Role of the Chaperones Hsp42 and Btn2 in Sorting Misfolded Intermediates to PACs in Different Budding Yeast Backgrounds

We next tested which of these chaperones participated in the assembly or disassembly of PACs during stress recovery after mild heat shock, or upon degradation of the misfolding intermediates (Appendix A). In budding yeast, many chaperones have been implicated in spatial protein quality control, with the aggregases Btn2 and Hsp42 having special relevance in the formation of transient foci [14,15,40]. To decipher whether they are sorting factors of the PACs monitored in the absence of inhibitors during mild stress (Figure 3a), we expressed Rho1.C17R-GFP in cells lacking Btn2, Hsp42 or both. As shown in Figure 3b, the lack of both Btn2 and Hsp42 totally (background W303) or partially (background BY4741) blocked the formation of these foci. In the case of the reporter Guk1-9-GFP expressed in W303 cells, some nuclear foci could still be detected in cells lacking both chaperones (Figure 3c). Based on Rho1.C17R-GFP, lack of Btn2 seemed to be sufficient to block PAC formation in both the cytosol and the nucleus in the W303 background (Appendix A), while individual deletions of *btn2* or *hsp42* could not avoid the presence of nuclear PACs in BY4741 (Appendix A). Quantification of the number of cells with PACs in the different genetic backgrounds is shown in Appendix A. Therefore, the requirement of Btn2 (for nuclear misfolded proteins) and Hsp42 (for cytosolic polypeptides) in PAC nucleation is not as strict as in the case of the Hsp40 Mas5 in fission yeast [23], but it is clear that both chaperones are required for PAC maintenance regardless of the reporter or the budding yeast background. Since Mas5 has proven to be an efficient sorting factor in *S. pombe*, we tried to suppress the defects of PAC formation in Δ*btn2* Δ*hsp42* cells by heterologous expression of Mas5, without success (Appendix A), suggesting that Mas5 cannot exert its PAC assembly functions in the absence of a fission yeast factor, probably its corresponding Hsp70, Ssa2.

### 2.3. The Disaggregase Hsp104 Participates in PAC Clearance during Heat Stress and Recovery

One goal of these PACs could be to maintain non-terminally misfolded proteins as reversible intermediates, for future refolding during stress recovery at non-stressful conditions (Figure 4a). Hsp104 has been proposed to participate in the disaggregation of heat-induced centers (for a review, see [41]). We expressed Rho1.C17R-GFP in cells lacking Hsp104 in W303 and BY4741 backgrounds. We shifted cultures of wild-type cells and cells lacking Hsp104 from 37 °C to 25 °C in the presence of CHX to inhibit protein synthesis. As shown in Figure 4b (W303 background) and Appendix A (BY4741 background), PACs assembled at 37 °C disappeared one hour after the shift to permissive temperature, and this disassembly required the presence of Hsp104. The disaggregation of PACs formed at 42 °C shifted to 25 °C also occurred at the same pace, as shown with Guk1-9-GFP (Appendix A). Therefore, Hsp104 is required to disentangle misfolded intermediates assembled into PACs when the temperature normalizes.

As shown in Figure 2a, Hsp104-GFP collapsed into PACs upon mild heat shock, and we tested whether it was required for PAC formation of the misfolding reporter. As shown in a W303 (Figure 4c) and a BY4741 (Appendix A) background, PACs were easily monitored upon stress in Δ*hsp104* cells, and in fact, these aggregation-like foci lasted longer than in wild-type cells during chronic heat shock. Thus, Hsp104 seems to have a role during stress recovery but also during sustained heat shock. In fact, the stability of the reporter expressed in a W303 background, which has a half-life of ~2 h in a wild-type background (Figure 1d), was enhanced in cells lacking Hsp104 (Figure 4d,e), suggesting that protein degradation is opposed by PAC formation, and requires the disaggregase Hsp104 to be initiated during prolonged heat shock.

### 2.4. Degradation Mediated by the Hsp40 Ydj1 Can Be Prevented by Formation of PACs

The Hsp40 Ydj1 is involved in the recognition and degradation of misfolded polypeptides, in combination with Hsp70 enzymes and specific E3 ligases such as the essential Rsp5 [42,43]. We tested how the lack of Ydj1 affects the homeostasis of our reporter, under unstressed and heat shock conditions (Figure 5a). As shown in Figure 5b,c and Appendix A, the concentration of the reporter was higher in cells lacking Ydj1, and the protein remained stable during the course of the experiment. Contrary to the total protein levels as determined by Western blot, the native form of Rho1.C17R-GFP, as determined by its apparent fluorescence, was lower in a Δ*ydj1* background, suggesting that the protein is improperly folded under basal conditions in the absence of this Hsp40 chaperone. Upon heat shock, PACs were still formed in Δ*ydj1* cells and lasted longer than in wild-type cells (Figure 5d and Appendix A). As previously reported, the aggregation foci were not as compact as in a wild-type background [18], although the weak fluorescence in Δ*ydj1* cells could explain this apparent difference. The *S. pombe* ortholog of Ydj1, Mas5, is required for PAC assembly but also connects physiological and heat shock triggered PQC. Thus, in physiological conditions, Mas5 contributes to the protein folding of thermo-sensitive proteins, and maintains them in their native conformation, avoiding misfolding and degradation [44,45]. The growth defects of cells lacking Ydj1, which were obvious even under permissive temperature, could be partially suppressed by the expression of Mas5 (Figure 5e; growth at 30 °C), suggesting that the fission yeast Hsp40 can support folding activities. Nevertheless, Mas5 did not contribute to the degradation of the misfolding reporter during mild heat stress, as shown by Western blot (Appendix A) and fluorescence microscopy (Figure 5d). Our experiments suggest that PAC formation is an early strategy during adaptation to mild heat shock to transiently shield misfolded proteins from Ydj1-mediated degradation. Ydj1 is not only required to disentangle polypeptides to send them to degradation if stress persists, but also during recovery to promote their refolding; as shown in Appendix A, while PACs disaggregate after one hour of growth at permissive temperature in a wild-type background, they did not in Δ*ydj1* cells 3 h after the shift. This finding is consistent with recent results supporting a Ydj1 role during protein disaggregation and reactivation [46].

To investigate the contribution of new protein synthesis on PAC assembly, we treated cells with CHX or puromycin during mild heat shock imposition. We formerly described that CHX inhibits PAC assembly in *S. pombe* since it freezes ribosomes and premature polypeptides which are required for PAC formation, while other inhibitors such as puromycin destabilizes polysomes and probably releases all translation components, enhancing PAC assembly [23]. As shown in Figure 6a (W303 background) and Appendix A (BY4741 background), the formation of PACs at 37 °C was fully abolished by treatment with CHX, whereas puromycin enhanced their formation (Appendix A). The degradation of Rho1.C17R-GFP, which occurred slowly in a wild-type background, was much faster if heat shock was imposed in the presence of CHX (Figure 6b and Appendix A); as expected, the addition of puromycin did not accelerate the kinetics of protein degradation (Appendix A). CHX-increased degradation was dependent, although not totally, on Ydj1 (Appendix A). As controls, the addition of CHX did neither result in the degradation of Rho1.C17R-GFP at 25 °C (Figure 6b and Appendix A), nor in the degradation of wild-type Rho1-GFP at 25 °C or 37 °C (Appendix A), confirming that degradation is exclusively triggered by protein misfolding after temperature upshift, and that the synthesis of our reporter did not greatly contribute to total protein concentration.

Another intervention reducing, although not fully eliminating, PAC formation, was depletion of the sorting factors Hsp42 and Btn2 (Figure 3 and Appendix A). Using two different reporters and in both genetic backgrounds (W303 and BY4741), degradation of the misfolding reporters was faster after stress imposition in Δ*btn2* Δ*hsp42* cells (Figure 6c and Appendix A).

In conclusion, our experiments suggest that there is an ordered strategy to adapt to mild heat stress in budding yeast (Figure 6d). Proteins prone to aggregate at 37 °C are sequestered into PACs in a chaperone-mediated manner. If heat stress persists (chronic heat shock), misfolded intermediates are disentangled from the foci by Hsp104 and Ydj1, and this Hsp40 mediates their degradation by the proteasome. Nevertheless, if stress stops (stress recovery), PACs can be disaggregated by Hsp104 and Ydj1 and transiently unfolded intermediates can be refolded to their native conformations, supporting survival and recovery.

### 2.5. Formation of Nucleolar Rings, NuRs, upon Severe Heat Shock Occurs in Both Yeasts and Is Chaperone-Mediated

We then characterized the fate of misfolded intermediates under severe heat stress, 42 °C, which halts cell growth in both budding (Figure 1b) and fission [23] yeasts. In *S. pombe*, a shift to 42 °C causes a faster and more pronounced coalescence of cytosolic PACs relative to 37 °C, also dependent on Mas5, at both the nucleus and the cytosol [23]. In budding yeast, upon severe heat stress, Rho1.C17R-GFP collapsed into cytosolic PACs in both genetic backgrounds (W303 and BY4741) (Appendix A), and Guk1-9-GFP also formed foci (Appendix A). The chaperones Btn2 and Hsp42 were not required for PAC nucleation at 42 °C (Appendix A), contrary to the essential role of Mas5 in PAC formation at 42 °C in fission yeast [23], suggesting that other redundant chaperones may contribute to the initial protein sorting towards PACs in budding yeast. Similar to the assembly of PACs at 37 °C, Btn2 and Hsp42 seem to have a role in the maintenance of PACs during severe heat shock (late time-points in Appendix A). The protein stability of the misfolding reporter Rho.C17R-GFP at 42 °C was even higher than during mild stress (compare the stabilities in a W303 background shown in Appendix A and Appendix A, suggesting that the composition, compactness and nature of the PACs at 42 °C shields proteins from the degradation fate.

As previously reported in fission yeast, severe stress is capable of triggering some nuclear aggregation-like foci termed nucleolar rings (NuRs) [47]. NuRs sequester essential nuclear factors and nuclear pore complex components, which return to their soluble conformation during temperature recovery. As shown in the insets of Figure 7a, misfolded intermediates also collapsed into NuRs upon severe heat shock both in fission and budding yeast. The nuclear localization of these structures was confirmed by co-expression with a nucleoporin marker (Appendix A). Similar to cytosolic PACs, Mas5 was essential for NuR formation, but the absence of Btn2 and Hsp42 did not block NuR assembly in both *S. cerevisiae* backgrounds. When we analyzed NuR composition in fission yeast, all the chaperones previously reported to be present at PACs formed at either 37 °C or 42 °C, namely Mas5, Ssa2, Ssa1 or Hsp104 [23], were also present at these nucleolar inclusions; as shown with the misfolding reporters, the absence of Mas5 also blocked the accumulation of Hsp104-GFP into NuRs (Figure 7b), and it also avoided the collapse of other nuclear/nucleolar components (Appendix A).

## 3. Discussion

Cellular strategies to survive to stressful situations are multiple and varied. We demonstrate here, using the same reporters in two yeasts and several genetic backgrounds, that formation of protein aggregation-like centers, PACs, is an early event during mild heat stress, 37 °C, which transiently shields unfolded intermediates from degradation; whether these polypeptides are later refolded or sent to degradation will depend on the duration of the stress. Under severe stress, 42 °C, PAC formation is faster and the number and location of the foci is different from those formed at mild conditions; in addition, nuclear proteins assemble into some localized structures around the nucleolus, NuRs, which include the misfolding reporters and the chaperones normally present at PACs. The PAC-mediated strategy to prevent misfolded protein degradation is particularly significant during recovery after stress imposition. Other reports propose a protective role of protein aggregation to promote cellular recovery upon return to normal conditions [26,27]. However, to the best of our knowledge, our recovery experiments are the first ones performed with mild heat shock as the only factor driving PAC assembly, without the putative perturbation of using genetic or chemical proteasome inhibition conditions to enhance PAC accumulation prior to recovery.

Another function proposed for protein aggregation during heat stress is to avoid the proteostasis collapse. Previous studies have shown that a high load of misfolded proteins can reduce the availability of the Hsp70 system and protein sequestration into PACs might neutralize this negative effect shielding possible binding sites of misfolded proteins for Hsp70 chaperones [24]. Both postulated functions, preventing proteostasis collapse and avoiding immediate protein degradation, can be complementary, the former being important to guarantee cell survival at the onset of the response while the second strategy becoming critical for recovery once the stress ends.

*S. pombe* and the budding yeast background BY4741 are not so different regarding the kinetics of PAC formation and disassembly. In both budding yeast derivatives, PACs are formed faster than in *S. pombe* (Figure 1c), but formation of PACs in the W303 background is more transient and protein degradation is an earlier event. We have measured mRNA levels of *ydj1* and *rsp5* in both backgrounds, BY4741 and W303, without observing differences. Another possibility to explain the differences on PAC disassembly could be the peak of expression of the chaperone Btn2. In budding yeast, PAC clearance seems to correlate with a decrease in Btn2 expression during mild heat stress [19,39]. It would be worthy to investigate whether the constitutive expression of Btn2 might prevent the disassembly of PACs and compare the expression profile of Btn2 during heat stress in BY4741 and W303 backgrounds. Although PACs labelled with the reporter Rho1.C17R-GFP dissolved at early time points of the incubation at 37 °C, at least two chaperones Hsp104 and Hsp42 remain at foci during prolonged heat stress (Figure 2), probably associated with endogenous misfolded substrates which are more resistant to Hsp104-mediated disaggregation. Furthermore, the apparent lag time of almost 60 min between the disappearance of Rho1.C17R-GFP from PACs by fluorescence microscopy in the W303 background (PACs are not visible at 60 min of heat shock; Figure 1c) and the start of its degradation by Western blot (2 h, Figure 1d) may also be explained by the loss of fluorescence of the GFP moiety after disaggregation, since Rho1.C17R-GFP could still be detected at 2 h of heat stress by immunofluorescence (Appendix A). Another interesting observation about the localization of aggregation-like foci is that there are no apparent nuclear PACs at 37 °C in *S. pombe,* even though this temperature has a similar effect in both yeast genus, barely altering growth curves ([23] and Figure 1b). We speculate that Btn2, which localizes at both cytosolic and nuclear PACs (Figure 2a), may be the sorting factor missing in fission yeast and capable of triggering nuclear PAC formation in these mild heat stress conditions.

Although the function of Btn2 has been mainly connected with the assembly of nuclear aggregates [19], our study demonstrates that Btn2 is also implicated in the formation of cytosolic PACs (Appendix A). In agreement with our observations, Btn2 was originally identified through its ability to cure cells of cytoplasmic amyloid formed by Ure3; both Btn2 and Ure3 were co-localized close to but outside of the nucleus [48]. Despite the clear participation of Btn2 and Hsp42 in PAC maintenance during mild stress [19], some nuclear foci can be observed in Δ*btn2* Δ*hsp42* cells depending on the budding yeast background (Figure 3b; BY4741 derivatives) or the misfolding reporter used (Figure 3c; Guk1-9-GFP in W303 background). Furthermore, foci formation upon severe stress, 42 °C, still occurs in cells lacking Btn2 and Hsp42, although cytosolic PACs may be slightly less apparent than in a wild-type background (Appendix A). On the contrary, the lack of the chaperone Mas5 of *S. pombe* fully avoids the formation of PACs at 37 °C and 42 °C, and NuRs are absent as well after severe stress [23]. We propose that other sorting factors in budding yeast may have redundant roles with Btn2 and Hsp42 and substitute them depending on the nature of the misfolding reporter, the extent of the heat stress or the genetic background.

We have shown here than both Ydj1 and Hsp104 are required for the disassembly of PACs, during chronic heat shock and during stress recovery (Figure 6d). In one case, the disentangled misfolding intermediates are sent to degradation, whereas they are refolded if the permissive temperature is restored. The E3 ubiquitin ligase Rsp5, which has been described as essential to degrade misfolded intermediates arising from heat stress, may be the sorting factor facilitating the degradation fate over the refolding one: Rsp5 only associates with the Hsp40 Ydj1 during heat shock [43].

Regarding the formation of NuRs, it was recently proposed in *S. pombe* that these structures, which appear upon severe heat stress, are composed of mRNA-related activities, and their recruitment into these rings may facilitate overall inhibition and facilitate irreversible damage, preserving them until favorable conditions return [47]. We show here that NuRs are also formed in *S. cerevisiae.* We also demonstrated that ring formation does not occur in *S. pombe* in the absence of Mas5, as shown using the misfolding reporter Rho1.C17R-GFP, the Hsp104-GFP chaperone and three other nuclear/nucleolar proteins (Figure 7b,c and Appendix A). We propose that nuclear proteins with a tendency to misfold are dragged into these foci in an Mas5-dependent manner, serving as a platform to condensate many other nuclear proteins whose function should be temporarily inhibited. Of note, the addition of translation inhibitors such as CHX does not fully prevent PAC or NuR formation at 42 °C [23], suggesting that pre-existing cytoplasmic and nuclear proteins have a higher tendency to misfold at this extreme temperature and collapse into foci with the aid of Mas5. In budding yeast, the presence of multiple and probably redundant factors supporting PAC formation does not allow us to test whether naturally misfolding polypeptides are also the seed of NuRs.

## 4. Materials and Methods

### 4.1. Growth Conditions, Yeast Strains and Plasmids

*S. cerevisiae* strains were grown in YPD or synthetic complete media lacking histidine (SD-his) with 2% glucose as described [49]. Origins and genotypes of strains and plasmids used in this study are outlined in Appendix A, respectively. Deletion mutant strains are from the Yeast EUROSCARF strain collection or generated by homologous recombination with PCR fragments from pFA6a plasmid derivatives [50]. *S. cerevisiae* chaperone and nuclear marker GFP-tagged strains are from the Yeast GFP Clone collection (ThermoFisher Scientific, Waltham, MA, USA). Strains expressing the different reporters (Rho1 WT-GFP, Rho1.C17R-GFP, Rho1.C17R-mCherry and Guk1-9-GFP) were obtained by transformation of W303 or BY4741 with linearized pRS403-derived integrative plasmids and selection for His+ strains or with pRS425 derived episomal (2 µ) plasmids and selection for Leu+ strains. Strains overexpressing *mas5* were obtained by transformation of the W303 derivatives with a linearized pRS404 derived integrative plasmid coding for *mas5* driven by the ADHi promoter and selection for Trp+ strains.

*S. pombe* strains were grown in minimal medium (MM) as described [51]. Strains expressing GFP-Mas5 and GFP-Ssa2 were obtained by transformation of HM123 with linearized integrative plasmids and selection for leu+ strains. Strain expressing Ssa1-GFP was obtained by homologous recombination with a PCR fragment from a pFA6a derivative.

### 4.2. Native and TCA Extracts and Western Blot

*S. cerevisiae* cells were grown to an OD_600_ of 0.5 and trichloroacetic acid (TCA) and native extracts were prepared as previously described [52]. Samples were separated by SDS-PAGE and detected by immunoblotting. GFP-tagged proteins were visualized with a monoclonal anti-GFP (JL-8, Takara Bio Inc., Kusatsu, Japan, #632381) and with either an anti-mouse HRP-conjugated (Amersham ECL Mouse IgG, HRP-linked whole Ab, Cytiva, Marlborough, MA, USA, #NA931) or a fluorescent secondary antibody (Goat Anti-Mouse IgG StarBright Blue 700, Bio-Rad, Hercules, CA, USA, #12004158). Ponceau staining was used as the loading control.

### 4.3. Fluorescence Microscopy

To visualize PACs at 37 °C and 42 °C, both *S. pombe* and *S. cerevisiae* cells growing in MM were harvested by centrifugation 1 min at 3000 and 7000 rpm, respectively. To visualize NuRs at 42 °C, *S. pombe* and *S. cerevisiae* BY4741 derived strains were processed as before, while W303-derived strains were previously concentrated 50-fold by centrifugation 1 min at 7000 rpm, resuspended in 1/50 of the original volume, and subjected to a 42 °C heat shock. All cells were visualized at room temperature as described before [23]. Briefly, images were acquired using a Nikon Eclipse 90i microscope equipped with differential interference contrast optics, a PLAN APO VC 100× 1.4 oil immersion objective, an ORCA-II-ERG camera (Hamamatsu, Herrsching am Ammersee, Germany), excitation and emission filters GFP-4050B and mCherry-C (Semrock, West Henrietta, NY, USA) or with a spinning-disk confocal microscope (IX-81; Olympus; Evolve camera, Plan Apochromat 100 Å~, 1.4 NA objective; Roper Scientific) and image acquisition software Metamorph 7.8.13 (Gataca Systems, Massy, France). Processing of all images was performed using Fiji (ImageJ, National Institutes of Health) [53].

### 4.4. Sensitivity and Survival Assays on Plates

For heat sensitivity assays, cells were grown in YPD to logarithmic phase, serially diluted, plated on YPD solid plates and allowed to grow at 30 and 37 °C for 2 days. For heat stress survival, cells were grown to logarithmic phase in YPD, pre-treated or not for 30 min at 37 °C and heat shocked at 50 °C for 20 min. Cells were then serially diluted and plated on YPD solid plates and allowed to grow for 2 days at 25 °C.

### 4.5. Growth Curves

Yeast cells were grown in SD-his at 30 °C to an OD_600_ of 0.5 at and then diluted to OD_600_ 0.3. Cultures were then split into 3 flasks and allowed to grow at 30 °C, 37 °C and 42 °C for 24 h. OD_600_ was manually recorded with a spectrophotometer every hour for the first 9 h and at 24 h

### 4.6. Immunofluorescence Microscopy

For immunostaining, 5 × 10^7^ cells were fixed with 3.7% formaldehyde in phosphate-MgCl_2_ buffer (100 mM K_2_HPO_4_/KH_2_PO_4_, pH 6.5, 0.5 mM MgCl_2_) at 25 °C for 1 h. After two washes with phosphate-MgCl_2_ buffer and one wash with sorbitol–phosphate buffer (1.2 M sorbitol, 100 mM K_2_HPO_4_/KH_2_PO_4_, pH 6.5), spheroplasts were obtained by treating cells with 2.5 mg/mL 20T zymolyase (Amsbio, Abingdon, UK, #120491-1) for 15 min at 30 °C in sorbitol–phosphate buffer with 20 mM ß-mercaptoethanol and washed three times with sorbitol–phosphate buffer. 20 µL of spheroplasts were spotted on poly-L-lysine (Sigma, Darmstadt, Germany, #P8920) coated multiwell slides and permeabilized by washing three times with 20 µL of 1% Triton X-100 in 100 mM K_2_HPO_4_/KH_2_PO_4_, pH 6.5. After blocking for 1 h in blocking buffer (20 µL of 1% BSA, 100 mM K_2_HPO_4_/KH_2_PO_4_, pH 6.5), anti-GFP (JL-8, Takara, Bio Inc., Kusatsu, Japan, #632381) was added at 1:200 in 20 µL of the same blocking buffer and left overnight at 25 °C. After four 5 min washes with blocking buffer, spheroplasts were incubated for 7 h with a 1:500 of the secondary antibody Alexa Fluor 555 goat anti-mouse IgG (Life Technologies, Carlsbad, CA; USA, #A21424) in 20 µL of blocking buffer, washed four times for 5 min with blocking buffer as above and finally embedded in 50% glycerol.

### 4.7. Quantification and Statistical Analysis

All experiments were performed at least three times and representative experiments are shown. Western blots developed with fluorescent secondary antibody and Ponceau-stained membranes were imaged with a Chemidoc^TM^ Imaging System (Bio-Rad, Hercules, CA, USA) and quantified using ImageLab v6.0.1 (Bio-Rad, Hercules, CA, USA). In the quantification graphs, bar heights represent the mean values and the error bars SD.

## 5. Conclusions

In this study, we have expressed in budding yeast the same misfolding reporters, derived from mutated *S. pombe* and *S. cerevisiae* endogenous proteins, previously shown to assemble into protein aggregate-like centers in *Schizosaccharomyces pombe* in experiments lacking proteasome or translation inhibitors. We have demonstrated that mild heat shock triggers reversible PAC formation, with the collapse of both reporters and chaperones in a process largely mediated by chaperones. This assembly postpones proteasomal degradation of the misfolding reporters, and their Hsp104-dependent disassembly occurs during stress recovery. Severe heat shock induces formation of cytosolic PACs, but also of nuclear structures resembling nucleolar rings, NuRs, presumably to halt nuclear functions. Our study demonstrates that these distantly related yeasts use very similar strategies to adapt and survive to mild and severe heat shock.

## Figures and Tables

**Figure 1 ijms-24-11202-f001:**
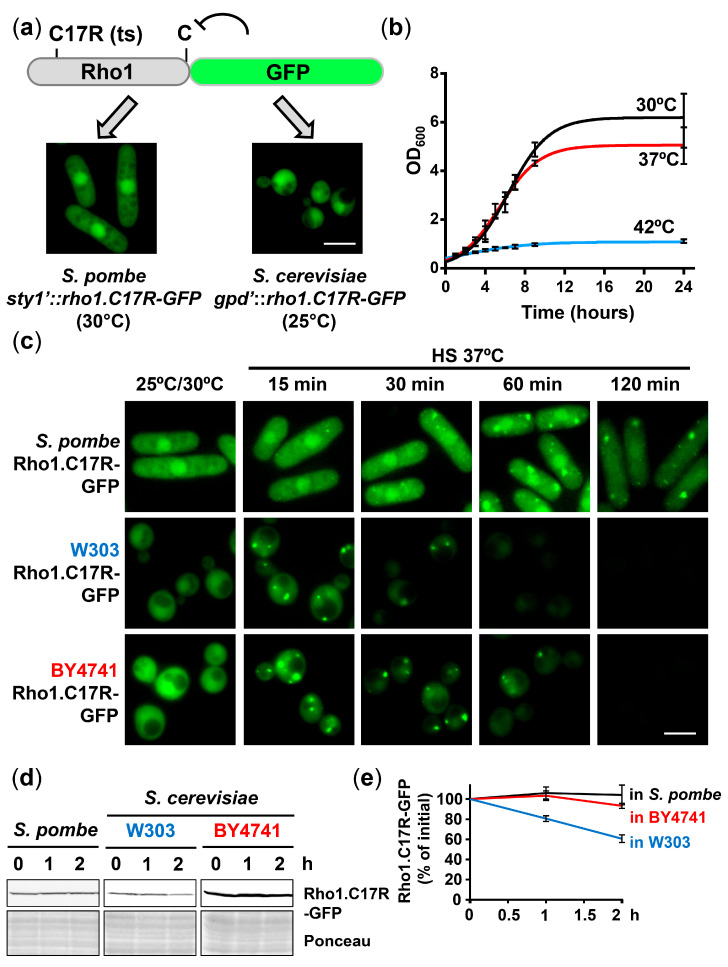
Protein aggregate centers (PACs) are assembled in budding yeast after moderate heat shock. (**a**) Expression of a temperature-sensitive reporter in yeast models. Scheme depicting Rho1.C17R-GFP construct, C-terminal GFP tagging abolishes geranylgeranylation of Rho1 protein. Expression of Rho1.C17R-GFP was under the control of *sty1* promoter in fission yeast and *gpd* promoter in budding yeast. To obtain similar reporter levels in both yeasts, growth temperature was 30 °C for fission yeast and 25 °C for budding yeast. (**b**) Only extreme temperature upshifts affect cell viability in budding yeast. W303 cells expressing Rho1.C17R-GFP were incubated at the indicated temperatures and cell growth was monitored measuring OD_600_. Graphs represent the nonlinear regression fit of three biological replicates. Error bars correspond to the SD (standard deviation) of the experimental data. (**c**) The thermo-sensitive mutant Rho1.C17R-GFP concentrates at PACs upon heat shock at 37 °C. Distribution of Rho1.C17R-GFP was analyzed by fluorescence microscopy at the indicated time points and temperature conditions. Wild-type fission yeast (strain 972) and two strains of budding yeast, W303 and BY4741, were included in this study. (**d**) Degradation of the reporter Rho1.C17R-GFP correlates with PAC clearance. Cells expressing Rho1.C17R-GFP were incubated at 37 °C for the indicated time points. Levels of Rho1.C17R-GFP were examined in TCA extracts by immunoblotting with anti-GFP antibody. Ponceau staining was used as the loading control. (**e**) Abundance of Rho1.C17R-GFP was quantified during heat shock at 37 °C. Graph represents the mean and SD of Rho1.C17R-GFP levels (% of initial) from three independent experiments. Scale bar, 5 μm.

**Figure 2 ijms-24-11202-f002:**
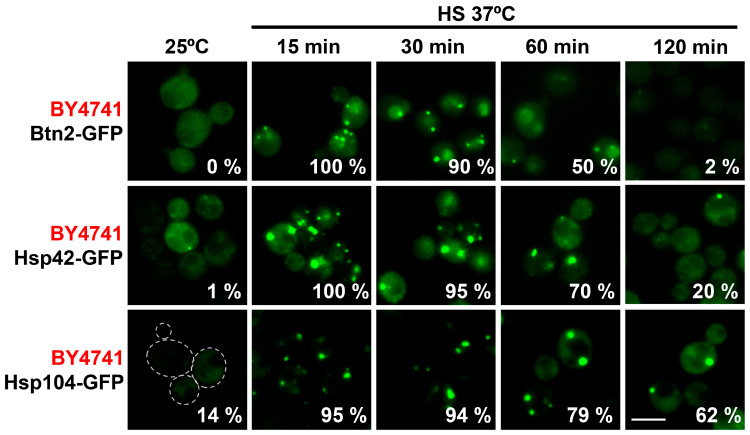
Different heat shock-induced proteins are present at PACs during mild heat stress. The small heat shock proteins Hsp42 and Hsp104 remain longer than Btn2 at PACs. Localization of different GFP-tagged chaperones was analyzed by fluorescence microscopy after incubation at 37 °C for the indicated times. The percentage of cells with PACs is indicated. Dashed lines indicate the borders of yeast cells. Scale bar, 5 μm.

**Figure 3 ijms-24-11202-f003:**
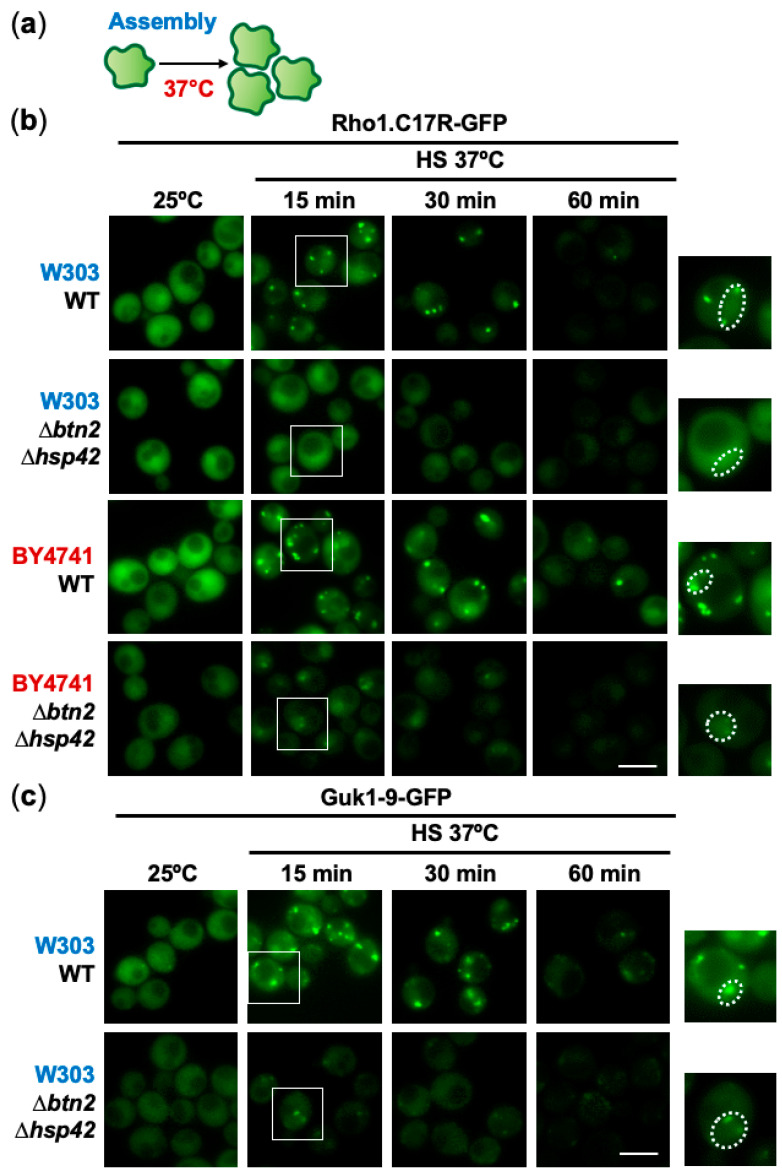
Btn2 and Hsp42 are required for PAC maintenance during moderate heat stress. (**a**) Assembly of PACs upon heat shock at 37 °C can be monitored using a thermo-sensitive reporter. (**b**) The absence of the chaperones Btn2 and Hsp42 impairs PAC formation and stability. WT and Δ*btn2* Δ*hsp42* cells expressing Rho1.C17R-GFP were heat shocked at 37 °C for the indicated time points and analyzed by fluorescence microscopy. Two backgrounds of budding yeast, W303 and BY4741, were used in this assay. Insets show a magnified region with one cell. Dashed lines indicate the borders of the nuclei, which were determined based on the stronger nuclear fluorescence of the misfolding reporter. (**c**) Btn2 and Hsp42 are also essential for the stability of PACs labelled with the misfolding reporter Guk1-9. Localization of Guk1-9-GFP was examined by fluorescence microscopy in WT and Δ*btn2* Δ*hsp42* cells after growth at 37 °C. Insets show a magnified region with one cell. Dashed lines indicate the borders of the nuclei. Scale bar, 5 μm.

**Figure 4 ijms-24-11202-f004:**
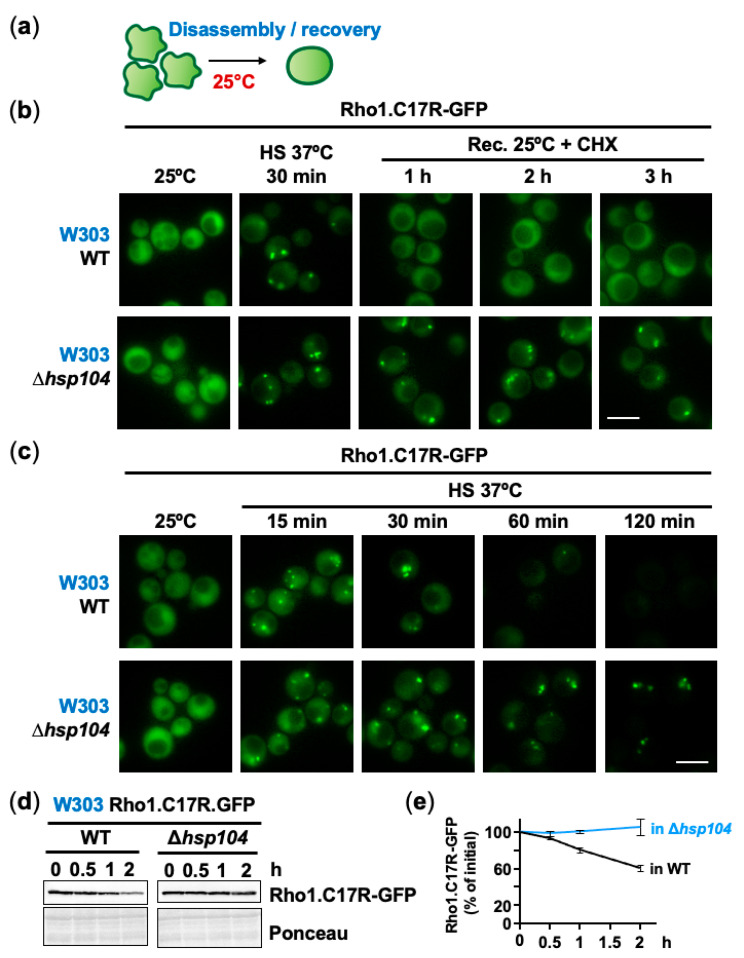
Hsp104 acts as disaggregase during heat stress and recovery at permissive temperature. (**a**) Incubation at 25 °C triggers disassembly of heat-induced PACs. (**b**) The activity of Hsp104 promotes PAC clearance once heat shock ends. WT and Δ*hsp104* cells carrying Rho1.C17R-GFP were heat shocked at 37 °C for 30 min and then incubated at 25 °C in the presence of CHX to block new protein synthesis. PAC dissolution was monitored at the indicated time periods by fluorescence microscopy. (**c**) PAC disassembly during heat stress also depends on Hsp104. The stability of PACs during a 2 h interval was analyzed by fluorescence microscopy in WT and Δ*hsp104* strains expressing Rho1.C17R-GFP. (**d**) Protein degradation is delayed in cells lacking Hsp104, where PAC disassembly is blocked. WT and Δ*hsp104* cells were heat shocked at 37 °C for the indicated time points and Rho1.C17R-GFP levels were monitored by immunoblotting with anti-GFP antibody. Ponceau signal was used as the loading control. (**e**) Quantification of Rho1.C17R-GFP concentration during upshift at 37 °C in WT and Δ*hsp104* cells. Graph represents the mean and SD of Rho1.C17R-GFP levels (% of initial) from three independent experiments. Scale bar, 5 μm.

**Figure 5 ijms-24-11202-f005:**
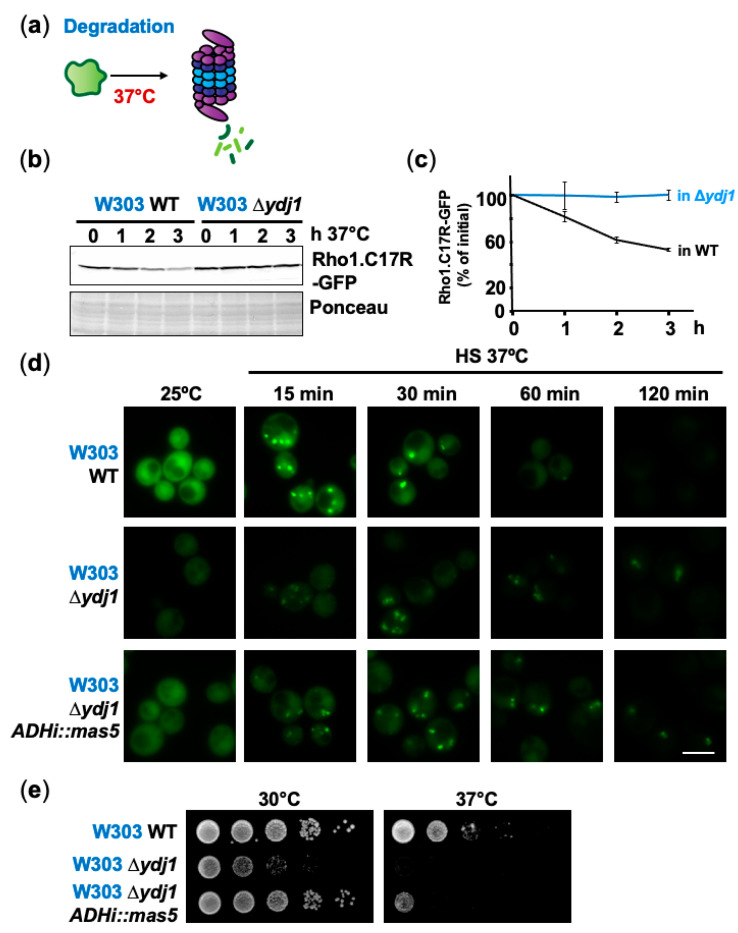
The Hsp40 chaperone Ydj1 participates in protein degradation during prolonged mild heat stress. (**a**) Proteins, not protected by PACs, are sent to the proteasome for degradation. (**b**) Degradation of misfolded proteins is blocked in cells lacking the chaperone Ydj1. WT and Δ*ydj1* cells expressing Rho1.C17R-GFP were heat shocked at 37 °C for the indicated time periods and the levels of misfolding reporter were detected by immunoblotting using anti-GFP antibody. Ponceau staining was used as the loading control. (**c**) The levels of Rho1.C17R-GFP were quantified during heat stress at 37 °C in cells treated as in (**b**). Graph shows the mean and SD of Rho1.C17R-GFP levels (% of initial) from three independent experiments. (**d**) Lack of Ydj1 delays PAC clearance during heat shock at 37 °C. Microscopy analysis of WT and Δ*ydj1* cells carrying Rho1.C17R-GFP after heat shock at 37 °C for the indicated times. Expression of Mas5 did not compensate for the absence of the chaperone Ydj1. (**e**) Mas5 expression in Δ*ydj1* cells increases cell viability at 37 °C. Serial dilutions of WT, Δ*ydj1* and Δ*ydj1 adhi::mas5* cells were spotted onto plates and grown at 30 °C and 37 °C to monitor cell survival. Scale bar, 5 μm.

**Figure 6 ijms-24-11202-f006:**
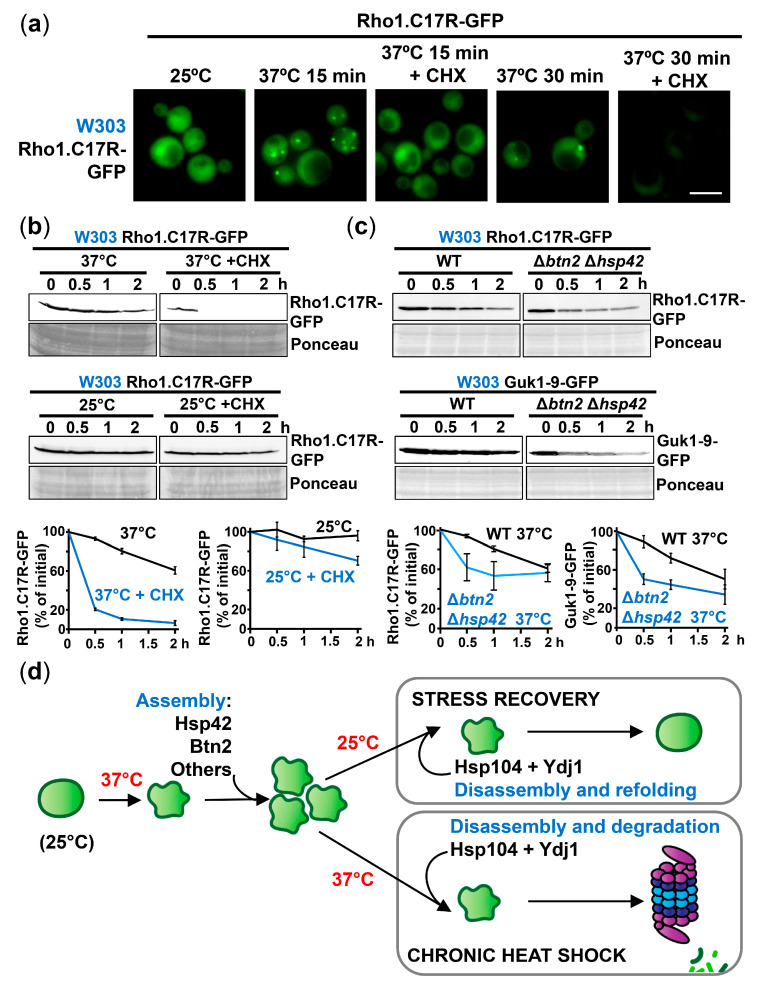
Protein sequestration into PACs postpones their degradation by the proteasome. (**a**) CHX treatment abolishes PAC formation upon mild heat shock. Cells expressing Rho1.C17R-GFP were examined by fluorescence microscopy after incubation at 37 °C for 15 or 30 min. Where indicated, CHX (100 μg/mL) was added to inhibit protein synthesis. (**b**) CHX-increased protein degradation is clearly observed after incubation at 37 °C, but not at 25 °C. The abundance of Rho1.C17R-GFP was monitored by immunoblotting after incubation at 25 °C or 37 °C, in the presence or not of CHX. Graphs show mean and SD of Rho1.C17R-GFP levels (% initial) from three independent experiments. (**c**) Protein degradation is accelerated in Δ*btn2* Δ*hsp42* cells exhibiting less PAC assembly. WT and Δ*btn2* Δ*hsp42* cells expressing Rho1.C17R-GFP and Guk1-9-GFP were heat shocked at 37 °C for the indicated times. The concentration of both reporters was analyzed by immunoblotting using anti-GFP antibody. Graphs represent the quantification of Rho1.C17R-GFP and Guk1-9-GFP levels (% initial) from three independent experiments. (**d**) Scheme illustrating the function of different chaperones during heat shock and stress recovery. Upon mild temperature upshift, both chaperones, Hsp42 and Btn2, participate in the assembly and maintenance of PACs. If permissive temperature is restored, misfolded proteins are extracted from PACs by the combined action of Hsp104 and Ydj1 and refolded. If the stress persists, misfolded proteins are also disaggregated but targeted to the proteasome for destruction. Ponceau staining was included as the loading control. Scale bar, 5 μm.

**Figure 7 ijms-24-11202-f007:**
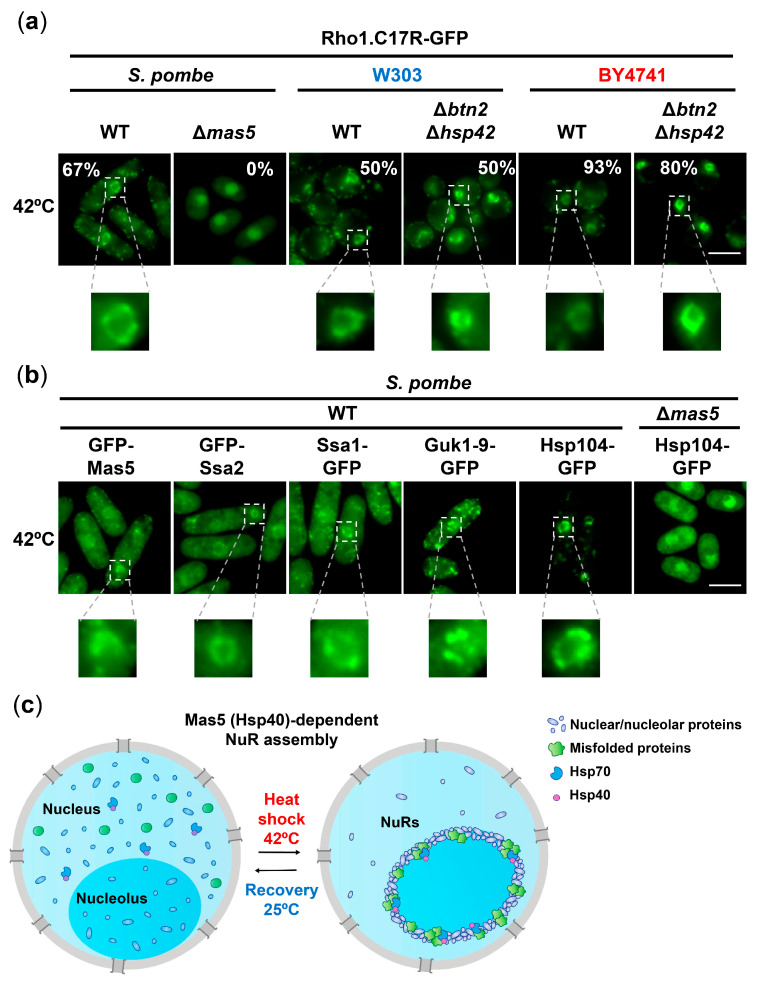
Nucleolar ring structures (NuRs) formed at 42 °C contain misfolded proteins and chaperones. (**a**) Lack of Mas5, but not of Btn2 or Hsp42, inhibits NuRs formation in yeast models. Microscopy analysis of WT, Δ*btn2* Δ*hsp42* and Δ*mas5* cells expressing Rho1.C17R-GFP after incubation at 42 °C for 15 min. The percentage of cells with NuRs is indicated. Insets show a magnified region of nuclei containing NuRs. (**b**) Several chaperones are recruited to NuRs upon severe heat shock in fission yeast. Localization of different GFP-tagged chaperones (Mas5, Ssa2, Ssa1 and Hsp104) and the reporter Guk1-9 was monitored by fluorescence microscopy after incubation at 42 °C for 15 min. Deletion of Mas5 also abolishes assembly of NuRs labelled with Hsp104-GFP. Insets show a magnified region of nuclei containing NuRs. (**c**) Scheme illustrating the assembly and disassembly of NuRs during heat shock and stress recovery. Upon a severe temperature upshift, Hsp70/40 chaperones, misfolded nuclear proteins and nuclear/nucleolar proteins are assembled into NuRs. If permissive temperature is restored, misfolded proteins are extracted from NuRs. Scale bar, 5 μm.

## Data Availability

The data underlying this article are available in the article and in its online Appendix A. All images included in the main and supplemental figures are available as Mendeley dataset (https://data.mendeley.com/datasets/3kcj6bjzvv/draft?a=31495303-2131-4e15-b5b2-f94c89c18d98) (accessed on 3 July 2023).

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
