# Peer review of "Formation of Transient Protein Aggregate-like Centers Is a General Strategy Postponing Degradation of Misfolded Intermediates"

_ijms, 2023, doi:10.3390/ijms241311202_

Round 1
Reviewer 1 Report
Boronat et al use unstable reporter proteins tagged with GPF to monitor the reaction of the yeast proteome to an upward shift in temperature. As the authors point out most papers focused on this phenomenon use artificially harsh conditions. In contrast Boronat et al perform most of their experiments under physiological relevant conditions. The authors observe that upon increase in temperature the reporter proteins form foci in the cells. Foci formation of proteins when cells are exposed to unfavorable conditions has been observed in the past (for instance: Narayanaswamy et al PNAS 2009; Widespread reorganization of metabolic enzymes into reversible assemblies upon nutrient starvation). Prolonged exposure to elevated temperatures results in clearing of the foci in a process that depends on the expression of chaperone proteins. Several chaperone proteins show a similar behavior but overlap of foci formed by the reporter and chaperones was not investigated. Foci formed by the reporter are both located in the cytoplasm as well as in the nucleus. When cells are shifted to 37C in the presence of cycloheximide foci don’t form and the reporter protein is rapidly degraded (Figure 6). What puzzles me is the discrepancy between the fluorescence signals and the protein signals. In Figure 1 fluorescence signals in S. cerevisiae cells show rapid foci formation and then decline to become nearly unobservable after 120 minutes. However, at the same 120-minute time point, protein levels have remained the same (BY4741) or are reduced by less than half (W303). Cytosolic pH levels should not drop to levels that would result in quenching of the GFP signal. What could be the explanation? In contrast in Figure 6 fluorescence is still clearly observable after the cells have been shifted to 37C in the presence of cycloheximide but protein levels have dropped below the detection limit of the blot that is shown. These results make it hard to justify the claims made about protein degradation in the paper.
Other comments.
It seems that both the Rho1 and Guk1 reporters produces a brighter signal in By4741 than in W303 (compare 25C time point in figure 1C and S1C. This could result in the signal seeming to diminish faster in W303 than in BY4741.
Figure 1B shows that the experimentalist went home after 9 hours. The deduced growth curve after the 9-hour point for cells grown at 30C could be represented with a dashed line. The sharp bend of the growth curve looks unrealistic.
In Figure 2 it seems as if Hsp104 is not (or poorly) expressed at normal growth conditions. Although Hsp104 is a heat shock protein there is plenty of it around when cells grow at optimal temperatures. This reporter also has the tendency to form foci under optimal growth conditions. Some statistics of cells with foci might be in place.
Figure 3 and 7. It might be informative to tell the readers in the figure legend how the position of the nuclei was determined.
In Figure 2 it seems that Btn2 levels (as indicated by fluorescence) are lower after 120 minutes at 37C than when the experiment was started at 25C. Btn2 levels are very low under optimal growth conditions as the protein is rapidly degraded by the proteasome. It seems unlikely that levels are indeed as suggested in this figure.
Figure 5 shows a very faint fluorescence signal in cells lacking YDJ1. However, protein levels at the zero-time point are similar to that of the WT.
Figure 7. How many cells show nuclear speckles? Could some statistics be provided?
Lines 470-472. Btn2p was identified by Kryndushin et al in 2008 (EMBO J. 27:2725-35) through its ability to cure cells of cytoplasmic amyloid formed by Ure2p. Btn2p and Ure2p amyloid showed colocalization close to, but outside of, the nucleus.
Line 541. Was the time between growth and microscopic observation constant? There have been situations where there was a huge gap between the two.
Reviewer 2 Report
This is a very high quality manuscript with exceptionally clear data. The subject is of general interest and the experiments are well designed and appropriately controlled and interpreted. I have a few suggestions that I think would be worth addressing, and should not require much time/effort.
“The authors state that “the requirement of Btn2 (for nuclear misfolded proteins) and Hsp42 (for cytosolic polypeptides) in PAC nucleation is not as strict as in the case of the Hsp40 Mas5 in fission yeast [23]”, but the cited study did not rule out that in fission yeast there may also be variation in strain backgrounds. In other words, perhaps in some S. pombe strains, the requirement for Btn2 is relaxed, similar to the case in S. cerevisiae. This is a small point but since one aspect of this work is background variation within a species, it would be important to not overstate what has been shown and can be concluded in S. pombe.
Lines 288-290: “the native form of Rho1.C17R-GFP, as determined by its apparent fluorescence, was lower in a Δydj1 background”: I see what the authors mean about the discrepancy between the levels according to immunoblot and the levels according to fluorescence in WT vs ydj1∆ cells, but I’m not sure how this reflects on the “native” form of the reporter. Is the idea that the GFP part of the fusion protein does not as efficiently fold/mature in the absence of Ydj1, hence lower fluorescence despite the same amount of total protein? If so, then the reduced signal may have nothing to do with the folding state of the Rho1.C17R part of the fusion protein. I was under the impression that GFP folding is largely independent of chaperones. Could there be something fundamentally different about the cytosol in ydj1∆ cells (after all, Ydj1 is involved in MANY different cellular processes) that reduces GFP fluorescence, apart from folding per se? An easy way to address this would be to simply express free GFP and compare the levels of fluorescence (and total protein) in WT vs ydj1∆.
Are the apparent NuRs really rings within, not around, the budding yeast nucleus? It’s hard to tell from the images provided. The authors could easily address this question by co-expressing a tagged histone (e.g., Htb1, as in Suppl. Fig. S1A), for example.
In terms of the differences between BY4741 and W303, which trait is dominant, and is the difference due to a small number of genes? An easy way to address these questions would be to simply make a diploid strain by mating an appropriate mutant of BY4741 (e.g. btn2∆ hsp42∆) with the same mutant W303alpha and assessing reporter dynamics in the diploid cells. Then, sporulate the diploid, dissect tetrads, and monitor reporter dynamics in the resulting haploid clones, and attempt to discern a segregation pattern. Does the W303-like trait segregate 2:2, for example, indicative of a difference in a single gene being responsible?
Given the authors’ (fully justified) emphasis in the Introduction on the nature of the reporters used in the literature, I would like to have more information here in the text about the reporters used in this study. Is anything known at the molecular level about the folding/aggregation state of the mutant proteins at different temperatures? Where do the responsible amino acid substitutions lie in the 3D structure of the proteins? Are they exposed on the surface vs buried in the hydrophobic core? Are these proteins normally oligomeric, in their native forms? Are the affected residues at an oligomerization interface? If no experimental structures are available, AlphaFold structures can be used.
Similarly, from the perspective of how misfolded proteins avoid chaperone engagement, I think it would be worth citing and discussing this new study: https://www.nature.com/articles/s41467-023-38962-z
Fig. S5A: It seems to my eye that the amount of the reporter increases at least 2-fold by 3 hr at 37˚C in the ydj1∆ cells, yet the quantification suggests that the levels are unchanged. How can this be explained?
Minor comments:
Figure 4b: there seems to be plenty of room to write out “Recovery” rather than the abbreviation “Rec.”, which I didn’t immediately recognize and is not specified in the figure legend. Same for Figure S4A,B, S5D
Line 326: “since if freezes” should be “since it freezes”
Line 329: “exacerbating PAC assembly”: “exacerbate” means to make a problem worse. Do the authors mean “promotes PAC assembly”?
